# Effect of Microwave Heating on the Acrylamide Formation in Foods

**DOI:** 10.3390/molecules25184140

**Published:** 2020-09-10

**Authors:** Joanna Michalak, Marta Czarnowska-Kujawska, Joanna Klepacka, Elżbieta Gujska

**Affiliations:** Department of Commodity Science and Food Analysis, The Faculty of Food Sciences, University of Warmia and Mazury, 10-089 Olsztyn, Poland; marta.czarnowska@uwm.edu.pl (M.C.-K.); elka@uwm.edu.pl (E.G.)

**Keywords:** acrylamide, food safety, Maillard reactions, microwave heating, microwave processing techniques, household microwave ovens

## Abstract

Acrylamide (AA) is a neurotoxic and carcinogenic substance that has recently been discovered in food. One of the factors affecting its formation is the heat treatment method. This review discusses the microwave heating as one of the methods of thermal food processing and the influence of microwave radiation on the acrylamide formation in food. In addition, conventional and microwave heating were compared, especially the way they affect the AA formation in food. Available studies demonstrate differences in the mechanisms of microwave and conventional heating. These differences may be beneficial or detrimental depending on different processes. The published studies showed that microwave heating at a high power level can cause greater AA formation in products than conventional food heat treatment. The higher content of acrylamide in microwave-heated foods may be due to differences in its formation during microwave heating and conventional methods. At the same time, short exposure to microwaves (during blanching and thawing) at low power may even limit the formation of acrylamide during the final heat treatment. Considering the possible harmful effects of microwave heating on food quality (e.g., intensive formation of acrylamide), further research in this direction should be carried out.

## 1. Introduction

Thermal processes are used in the food industry to provide safe products with prolonged shelf-life. Baking, roasting, frying, sterilization, or microwave heating can affect food beneficially or negatively. Negative effects of thermal processing include the formation of compounds that do not occur naturally in foods and may be, inter alia, mutagenic, carcinogenic, or cytotoxic [1]. These compounds are referred to as process contaminants, that cannot be entirely avoided during food processing. However, technological treatment can be optimized in order to reduce the levels of their formation. The impact of chemical contaminants on consumer health and well-being could be noticeable after many years of prolonged exposure at low levels (e.g., cancer). The toxic compounds formed during the thermal processing of food products include, among others, heterocyclic aromatic amines, nitrosamines, polycyclic aromatic hydrocarbons, 5-hydroxymethylfurfural, furan, and acrylamide (AA) [1,2,3,4,5,6]. Special attention is paid to acrylamide, as this compound has been detected relatively recently. One of the most important challenge is to minimize the formation of this toxin both during industrial food-processing as well as food preparation by consumers. Home-cooking choices could have a substantial impact on the level of acrylamide. 

Thanks to technological development and the use of both industrial and home-cooking techniques, the knowledge of the application of the thermal treatment to achieve specific food qualities, has increased. Numerous studies have shown that thermal treatment changes the physical and chemical structure of food [1,7,8,9,10,11]. Moreover, it is believed that not only heating temperature and time are the most important factors in the formation of hazardous compounds in foods. The heat treatment methods can also be responsible for food toxicant formation (their type and level) [12,13,14,15,16,17,18,19]. In recent years, new technologies such as microwave heating have experienced increased popularity as alternatives to conventional processing methods having various applications in the food industry and foods prepared at home, by catering services or served in restaurants. The increased use of microwave techniques also results from the fact that it helps to overcome the disadvantages of conventional methods [14,20].

In this review, we will discuss the use of microwaves in the food industry and focus on the formation of acrylamide as a result of both microwave and conventional heating methods.

## 2. Structure and Properties of Acrylamide

Acrylamide (CH_2_=CH-CO-NH_2_) is a low molecular organic compound consisting of carbon (50.69%), hydrogen (7.09%), nitrogen (19.71%), and oxygen (22.51%), with a molecular weight of 71.08 g. This compound is polar and well-soluble in water, methanol, and ethanol. Since 1950, it has been synthesized on an industrial scale. Polyacrylamide is used in various industries and in agriculture [21]. Research conducted in 1990 proved its toxicity, which was confirmed in epidemiological studies. Its genotoxicity was demonstrated in studies on somatic and sex cells. In animal studies (mice and rats), it was observed that acrylamide induced the formation of cancer cells [3,4,7]. Only the monomer form of AA induced a toxic effect; in a polymerized form, no harmful effect on human or animal organisms was observed. After ingestion, acrylamide is rapidly absorbed and distributed in animals and humans throughout the whole body. It can be found in many organs—such as the thymus, liver, heart, brain, kidneys, as well as in human placenta and breast milk—thus making it easily transferable to fetus or newborn infants. Toxicokinetic studies on humans indicated, that nearly 60% of absorbed acrylamide is excreted in the urine (86%) and that unchanged acrylamide accounts only for 4.4% of the up-taken dose [3]. Acrylamide shows low reactivity with DNA. However, it is metabolized to a reactive epoxide metabolite, glycidamide, which in contrast to acrylamide gives rise to stable adducts to DNA, which can cause genetic mutations and damage chromosomes [3,21,22]. In 1994, the International Agency for Research on Cancer [23] classified acrylamide as a “potentially carcinogenic” compound (Group 2A). In the European Union classification system, acrylamide occurs in the second category as a carcinogen and a mutagen [24]. In 2010, the European Chemicals Agency [25] added acrylamide to the list of substances of a very high concern.

Before 2002, it was thought that acrylamide does not occur naturally, but only as a result of chemical synthesis. However, in April 2002, the Swedish State Food Agency and Stockholm University researchers published surprising data on the considerable acrylamide content in food (from 30 μg/kg to 2300 μg/kg), especially in potato and cereal products subjected to high heat treatment [26]. In the same year, acrylamide was added to the list of toxic substances present in food. Based on the available studies on acrylamide levels in foods and their intake, the World Health Organization estimated the average acrylamide intake from food at the level of 0.3–2.0 μg/kg b.w./day [27], and even several times higher for children due to their lower body weight [28]. Shortly after this, other world scientific centers reported even higher acrylamide contents (up to about 12,000 μg/kg) in starchy foods treated with high temperature—such as chips, potato chips, bread, especially crisps, biscuits, crackers, cookies, breakfast cereals, and others. This compound was also found in meats and fish subjected to high heat treatment, as well as in various types of fast foods and in confectionery, chocolate, and sweet snacks, also in cocoa, milk replacers, and baby gruels. Significant acrylamide content was also found in coffee. Over the last years, several studies have reported low levels of AA in milk after heat treatment and dairy products [3,4,29,30,31,32,33,34,35,36].

It is now recognized that the main route of acrylamide formation in food is through Maillard reaction [37] (Figure 1). Acrylamide is formed in heated foods by condensation of the amino group of an amino acid (asparagine) with the carbonyl group of sugar. The effect of this reaction can be an intermediate product—the Schiff base, that can be transformed into acrylamide in direct conversion, or as a result of subsequent steps involving a number of reactions which can form acrylamide [6]. Alternatively, AA can be formed as a result of Strecker degradation, which occurs in foods subjected to very high temperatures. So far, results of studies confirmed the mechanism of acrylamide formation and the key role of asparagine in this compound formation. The carbon skeleton of acrylamide is derived from asparagine, as confirmed by mass spectrometry using labelled carbon and nitrogen atoms [38]. It has also been shown that α-hydroxycarbonyl compounds, particularly fructose and glucose, are more reactive and much more efficient in converting asparagine to acrylamide than α-carbonyl compounds. The α-hydroxyl group plays a key role in the degradation of asparagine as a result of the reduction of total Maillard reaction activation energy. Carbon atoms derived from reducing sugar molecules are not used to form acrylamide, but only to support the conversion of asparagine to acrylamide [4,21,30,31,37,39].

Another precursor of acrylamide formation is acrolein, which is produced in oils during heating above smoke point. At such a high temperature, triacylglycerols are hydrolyzed to glycerol, from which acrolein is formed as a result of dehydration. Acrolein further oxidizes to acrylic acid and produces acrylamide in the presence of asparagine. In this process, acrylic acid provides the carbon source and asparagine supplies the amino group. Subsequently, the peroxy radicals can initiate the polymerization of acrylamide in the presence of oxygen [3,6,22].

Acrylamide (AA) can be formed in heated food not only in Maillard reactions and transformations of acrolein, but also by the conversion of bacrylic acid, wheat gluten, or by de-amination of 3-aminopropionamide or as a result of asparagine enzymatic decarboxylation [3,30,31,37,40]. However, Maillard reactions remain the main route for acrylamide formation as the result of a non-reactive matrix (starch or protein) presence.

Major determinants of acrylamide formation in food are asparagine and reducing sugars or reactive carbonyls. Their content depends primarily on the species and variety characteristics of food products, methods of their cultivation, harvesting, and storage [15,17,41,42,43]. For example, in wheat flour, acrylamide formation is determined by asparagine level which accumulation increases dramatically in response to sulfur deprivation and, to a much lesser extent, with nitrogen feeding. In potatoes, in which sugar concentrations are much lower, the relationships between acrylamide and its precursors are more complex. However, the level of asparagine as a proportion of the total free amino acid pool has been shown to be a key parameter indicating that, when sugar levels are limiting, competition between asparagine and the other amino acids for participation in the Maillard reaction determines acrylamide formation [28,41,42,43]. In addition, an important factor affecting the AA level in the product is high temperature (notably above 120 °C) [41]. The highest amount of acrylamide was found in foods heated above 160–180 °C [6,15,16]. On the other hand, long-term heating of foodstuffs at higher temperatures, especially above 200 °C, can promote the degradation of acrylamide [3,37]. Careful choice of process parameters can be used as an effective tool to reduce AA content in thermally processed foods [6]. It was also found that low humidity (10–20%) of potato products and of cereal products (less than 10%), intensified the formation of this compound [16]. The results of many studies indicate that the formation of acrylamide in foods is limited with a water activity above 0.8 and below 0.4, and the optimum level of water activity is approximately 0.4 [6]. The formation of AA in foods is most conducive at pH 7–8, therefore a change in the pH of processed food may lead to a decrease in the content of this compound. This observation was used by Salazar et al. [42], who indicated that addition of lime to tortilla chips decreased the AA content by 36–52%. This may be due to the fact that lime concentration can play a vital role in mitigating acrylamide during nixtamalization through inhibition of Schiff Base. The addition of other food products may also lead to a decrease in this compound content. Jing et al. [40] analyzed the effect of extracts obtained from buckwheat seeds and sprouts on acrylamide formation and the quality of bread. The authors indicated that these extracts reduced the AA content of 16.7–27.3%, depending on the kind of extract. The authors also demonstrated significant positive correlations between total phenolic compounds content, antioxidant activity of the extracts, and reduction in the acrylamide level. Despite many studies and extensive knowledge about the mechanism of acrylamide formation in food and the factors influencing it, there are still many uncertainties [3,21,30,31,43,44,45].

Numerous studies have found that the acrylamide content in foods varies widely, on average less than 100 μg/kg, and in extreme cases from even below 10 μg/kg in high protein products, up to 100–4000 g/kg in products with high sugar content. The highest content of this compound was determined in foods subjected to thermal processes such as potato frying and roasting, cocoa and coffee roasting, bread and pastry, and cereal heat treatment [3,16,30,31,32,46] (Table 1).

Since 2002, many European countries have monitored AA in food. Due to the presence of this compound in many food products and its potential negative health effects, the European Commission issued a Recommendation [51] on the monitoring of acrylamide levels in food. Based on the results of the 2007–2009 study, the Commission issued a Recommendation in January 2011 on the “indicative values” of acrylamide content in foods, which are the main source of this compound in the diet [52]. These values do not constitute safety thresholds. If they are exceeded, they should point to the need to investigate the appropriate tools to reduce the AA content in these products. In this way, the risks associated with exposure to acrylamide in food should be controlled and regulated. The “indicative values” set out in the Commission Recommendation from 2013 [53] are now replaced with the values in the Commission Regulation 2017 [54] and in 2019 a new regulation was published specifying monitoring methods of acrylamide content in those food products for which it was not previously indicated [55]. The reference values recommended in 2013 [53] and 2017 [54] have been lowered because monitoring has shown a decrease in acrylamide content in certain food categories [56,57].

So far, a number of different acrylamide content reduction strategies have been developed in the “Toolbox”. The “Toolbox” provides information on various ways to reduce AA and addresses four areas: agronomy, recipes, processing, and final preparation. The guidelines are regularly updated based on new research findings [37,57,58,59]. However, despite the efforts made in many countries to reduce acrylamide levels in food, its content has been reduced only slightly and only in some products such as potato products, and cereal-based foods for infants and young children. At the same time, there has been an upward trend in other food categories such as bakery products and coffee, which may indicate that all the tools proposed in the “Toolbox” are ineffective or unrealistic in reducing food acrylamide levels [57]. The data show how difficult it is to reduce the content of this compound in food and that its presence in many food products is a constant problem.

## 3. Characteristics and Application of Microwave Treatment versus Traditional Heating

Electromagnetic waves (microwaves) have been used since the 1940s. The food industry is the largest beneficent of microwave energy in such processes as drying, blanching, cooking, defrosting, pasteurization, sterilization, baking, heating, and fat melting. Microwave ovens are also widely used in everyday life for rapid food heating, particularly for so-called “convenient food” [15,20,35,60]. The use of microwaves for food processing is developing continuously world-wide. Faster heating and high energy efficiency are the major advantages of microwave processing of foods. However, there are still some problems in microwave processes in terms of food quality and non-uniform heating [20,60].

Microwaves are electromagnetic waves which frequency varies from 300 MHz to 300 GHz. The corresponding wavelengths span a range from 1 mm to 1 m, exhibiting the medial position of microwaves between infrared and radio waves. Domestic microwave appliances operate generally at a frequency of 2.45 GHz. Industrial microwave systems operate at frequencies of 915 MHz and 2.45 GHz in order to avoid interference with telecommunication devices [61,62,63].

Microwave heating is caused by the ability of the products to absorb microwave energy and convert it into heat. Microwave heating of foods mainly occurs due to dipolar and ionic mechanisms. When food is placed in a microwave oven, various food ingredients behave differently. The main ingredient that enables foods to be heated by microwaves is water. The higher the water content of food, the faster the heating rate is [62]. Except for water molecules, the polar particles of food ingredients are also subjected to intense electromagnetic field action resulting in the polar molecule rotation. This extremely high rotation rate causes water molecules and the polar particles of food ingredients to collide with each other at very fast rate. This creates friction between molecules and it generates heat. The heat flows through the food by conduction, convection, or radiation and therefore food warms up [63].

Microwave heating takes place not only on the surface of wet biological materials, but also within them. In conventional thermal processing, energy is transferred by conduction from the product surface to the inner part. This depends mainly on temperature gradient and the thermal conductivity of the product. Compared to conventional heating methods, microwave heating techniques have several advantages such as volumetric heating, high heating rates, and short processing times. In addition, they are convenient to operate and control, energy efficient, easy to install and clean-up, require a short start-up time, etc. [64]. These advantages make microwave ovens common household appliances today. Thus, the food industry has been developing microwavable products for families and foodservice industry. Microwave heating has been also successfully used in food industry, including tempering or thawing of bulk frozen foods (meat, fish, and others), cooking of bacon and sausage and drying of pasta and vegetables, and also in disinfesting of insects in agricultural commodities, blanching of vegetables, inactivating of enzymes, pasteurization of breads, preserving hams and sausage emulsions, and sterilization of food products. In summary, typical applications of microwaves include blanching, drying, thawing, tempering, pasteurization, sterilization, baking, and cooking [8,62,63,64,65,66].

### 3.1. Blanching

Blanching is a unit operation practiced in food industry and is carried out by immersing food materials in hot water, steam, boiling solutions containing acids and salts, or in a microwave applicator. This process is generally used for color retention and enzyme inactivation of vegetables and some fruits, prior to further processing (freezing, frying, drying, canning, or sterilization) [63]. The conventional blanching method is closely associated with the serious issues like loss of weight, leaching, and degradation of nutritive components such as sugar, vitamins, and minerals. The loss of nutrients is found to be lower for the microwave-blanched samples compared to the conventional blanching. The advantage of microwave blanching over conventional method includes also speed of operation, no additional water requirement, energy savings, precise process controls, and faster start-up and shut-down times [61,63,67,68]. However, since there is uneven distribution of moisture and ions in different parts of the food products, the microwave heating can lead to problems of non-uniform heating and uneven energy distribution causes hot and cold points in the product, which makes this method of heating more inhomogeneous.

### 3.2. Drying

The goal of food products drying is to increase their storage stability without changing physical and chemical composition. Drying is one of the most time- and energy-consuming process in the food industry. For this reason, new methods are intensively sought and one of them may be microwave drying. It is a complex process involving heat and mass transfer, which is based on the volumetric heating [20]. Vapor is generated inside a food item and transported toward the food material surface, creating a higher heat transfer and thus, a much faster temperature rise than in the conventional heating. Meanwhile, in the conventional drying, moisture is initially flashed off from the surface and the remaining water diffuses slowly to the surface [20,62,67]. The major disadvantage of microwave drying is difficulty in controlling the final product temperature. High temperature at high microwave power can greatly destroy the nutrients, especially heat sensitive components.

It was found that microwave energy combined with other drying methods, including the conventional drying, can improve the drying efficiency as well as the quality of food products [5,20,67]. Microwave assisted air drying is found to be helpful at the final stages of drying food products, especially fruits and vegetables. It increases the drying rate and enhances the rehydration capacity of dried products and also overcomes shrinkage problems [5,63,69].

### 3.3. Thawing and Tempering

One of the most successful applications of microwave in the food industry is microwave thawing and tempering. Frozen meat, fish, vegetables, fruit, butter, and juice concentrate are common raw materials for many food-manufacturing operations. Few processes can handle the frozen material and it is usually either thawed or tempered before further processing [67,70]. The major problem of the conventional thawing and tempering are large space requirements and long-time which may result in chemical and bacteriological deterioration. When frozen foods are thawed, the surface area of the food is the first to rise in temperature and bacterial multiplication can recommence which was restricted in frozen foods [60]. The use of microwave energy solves this problem, as heat is generated with the food from the inside to the surface, making microwave thawing a faster process than other methods [67]. The major disadvantage of the microwave thawing is still that it does not occur uniformly and this phenomenon is known as runaway heating. Microwave tempering has several advantages over most common thawing processes, among other things, it can handle large amounts of frozen product at small cost, has a high yield, and is accomplished in small spaces with no bacterial growth compared to conventional tempering techniques either with water or air. The process is successfully used by meat, fish, and poultry industries for further processing while the dairy industry exploits the technology for tempering of butter and frozen foods and to reduce the chances of rancidity during bulk freezing of butter [63,67].

### 3.4. Pasteurization and Sterilization

Pasteurization is the process that uses relatively mild heat treatment on foods to kill key pathogens and inactivate vegetative bacteria and enzymes to make food safe for consumption. Most frequently, milk and fresh fruit juices are pasteurized, where minimum process is necessary to eliminate health-associated hazards. However, the thermal treatment given does not kill bacterial spores, and hence the product is not stable at room temperature. Under refrigerated storage conditions, one can expect 2–6 weeks of shelf life. Sterilization is a more severe thermal treatment of foods. The process is designed to achieve commercial sterility of the products, giving it long-term shelf stability [63,71]. Traditional heat sterilization is mainly carried out by heating and is characterized by slow heat transfer and long sterilization time, which seriously affects the quality of food products. Therefore, many authors claim that microwave heating is preferred for pasteurization and sterilization over conventional heating.

Several theories were presented to explain how electromagnetic fields might kill microorganisms without heat. Some researchers have claimed non-thermal or enhanced thermal effects, to be associated with microwave heating on the destruction of microorganisms and inactivation of enzyme. Many researchers, however, rejected and continue to reject any molecular effects of electric fields compared with thermal energy [63,71,72,73,74,75]. According to U.S. Food and Drug Administration [76], the additional inactivation or non-thermal inactivation effect of microwave process on the destruction of microorganisms and inactivation of enzymes is inadequate in degree. Hence, when describing the inactivation kinetics of microorganisms by microwave heating, it is recommended to include only thermal effects in the model [76]. Despite the passage of years and many studies the issue still remains controversial. The effects of microwave and dielectric heating are clearly fields where there are knowledge gaps, and further studies are needed [63,71,72,73,74,75].

There are many limitations in the use of microwave pasteurization and sterilization in food industry. The results of many recent studies carried out by experts in the field of microwave technology applied to food indicate that continuous-flow microwaveable pasteurizers could be used for milk and juice processing. Microwave pasteurization of ready-to-eat meals has also been found to be a commercial success in European countries although US industries are still reluctant to adopt the technology [63,77]. However, replacement of the conventional heating by microwave energy source is not possible without understanding the real heating and inactivation mechanisms, temperature distribution in multilayered foods, and other critical factors. Both pasteurization and sterilization are based on time–temperature combination processes applied to food products to achieve intended target lethality [71]. The major drawback in the microwave sterilization is the lack of availability of actual temperature profiles. Measurement of temperatures at few locations does not guarantee the real temperature distribution of the product during microwave heating, as the heating pattern can be uneven and difficult to predict and change during the heating. Furthermore, it is not always true that the microwave-assisted process results in better quality retention of food products. The degradation kinetics of either quality, sensory, or nutrients depend upon many factors like nature of the food products, food geometry, dielectric properties, and oven designs as compared to conventional thermal processing [72,75]. The dielectric properties of the food product significantly vary during heat processing. These changes in dielectric properties could affect the heating pattern qualitatively, while such factors are not serious in conventional thermal processing. The studies demonstrated better results for microwave sterilization of food in terms of nutrient quality retention and microbial elimination in comparison with earlier results. However, the uniform temperature profile in the whole food product is not clearly confirmed [63,74].

### 3.5. Cooking and Baking

Cooking is one of the most familiar applications of microwave ovens. Microwave heating is so rapid that it takes the product to the desired temperature in a short time. Microwave ovens are well suited for cooking the food in small quantities, especially for households. Because of this, microwave ovens are commonplace in households and are established there as devices of everyday use. Their primary function is still the reheating of previously cooked or prepared meals. The popularity of these devices is constantly growing. Therefore, in recent years, food processors have developed many new-generation microwaveable foods with suitable packaging materials to meet the demands of a growing group of consumers. As demonstrated, the nutritious characteristics of the food retained in microwave cooking are quite good, but it does not attain the typical flavor of the cooked dish. Therefore, new combination techniques of microwave treatment with conventional technologies are recommended [63,78].

There are numerous reports on baking using microwaves [20,30,61,63,64,65,67,78]. In many cases, a comparison is made between microwave and traditional baking. Quality problems associated with microwave baking include reduced height of the product, dense or gummy texture, crumb hardness, and an undesirable moisture gradient in the final baked product. Conventional baking using hot air provides suitable color and texture. In microwave baking, a sufficient brown color on the surface and crust formation of breads or other food are not possible. During microwave heating, the air surrounding the food product is cold and water evaporating from food gets condensed on contact with cold air, which results in a lack of crispness of the food product. Other reasons are the differences between microwave and other heating mechanisms and specific interactions of each component in the formulation with microwave energy. The combination of microwaves with other heating systems is recommended to reduce processing time and increase the quality of products [63,64,79].

## 4. Acrylamide in Microwave Heating

The influence of conventional heating methods on the acrylamide formation in foods is relatively well known. Various factors affect the reaction yields of acrylamide, such as heat processing method, heating temperature, heating time, concentrations and types of sugar compounds, and water content. This main effect of temperature and time in heat processing methods on acrylamide formation has been reported by many researchers [4,6,12,15,35,80,81,82,83,84,85]. Some studies indicate that microwave processing can facilitate Maillard reactions as with conventional heating methods. This raises questions whether the acrylamide formation mechanism is similar to the traditional heating methods and which process (conventional or microwave) produces more acrylamide. So far, too little information is available on the effects of microwaves on the acrylamide formation and interactions between food ingredients. Some authors have suggested that more acrylamide may be formed under microwave heating comparing to conventional heating methods [15,35,84,85,86,87,88]. A possible explanation for this might be that microwaves offer fast temperature increase in the foods owing to their capacity to generate heat energy inside the food, without requiring any medium as vehicle for heat transfer. Products with low thermal conductivity may quickly reach high temperatures, and this does not occur in conventional heating. Therefore, microwave heating provides a favorable medium for the occurrence of acrylamide and probably significantly affects its formation and kinetics. Juodeikiene et al. [89] indicated 49.5–74.3% higher acrylamide content in corn products after vacuum microwave treatment compared to the infrared irradiation. Chen et al. [90] reported that microwave-puffed shrimp chips contained higher amounts of AA than deep-fried ones. Many studies concern the formation of large amounts of acrylamide in microwave heated potato products (Table 2). Michalak et al. [84] found that microwaving of frozen pre-prepared potato products, such as chips and wedges, led to higher levels of acrylamide in the final cooked product than any other cooking method. Hamid et al. [82] analyzed different ways of defrosting potato fries and indicated, that although the thawing conditions did not significantly affect the formation of acrylamide during frying, microwave thawing was the best method due to the (relatively) low acrylamide and oil contents in ready-to-eat products and their desirable color attributes. The authors also concluded that manufacturers of frozen par-fried potato strips should specify the use of a microwave for thawing as part of the frying instructions on the packaging to reduce acrylamide formation in French fries. Tareke et al. [33] found a large amount of acrylamide (551 μg/kg) in microwave-heated grated potatoes, even higher than in the same potato samples under a frying treatment (447 μg/kg). Takatsuki et al. [86] reported higher concentrations of AA in baked potato, asparagus, and green gram sprouts after being precooked by the microwave pre-treatment than in the products without precooking. The experiment conducted by Michalak et al. [15] showed that the mean AA content in croquettes heated in the microwave was significantly higher (420 μg/kg) than in baked (360 μg/kg), deep-fried (298 μg/kg), and pan-fried (285 μg/kg) samples. These studies have shown that the way the heat is transferred to food seems that it has a significant effect on the rate of AA formation and that microwave heating can cause more acrylamide formation in products than conventional food heating treatment.

On the other hand, some authors showed no acrylamide formation in foods under microwave heating [91,92,93]. Burch [93] found that fresh potatoes microwaved in their peel contained negligible levels of acrylamide. Sansano et al. [94] indicated that microwave frying resulted in AA reduction ranging from 37% to 83% compared to deep-oil frying. Asadi et al. [12] showed, that microwave roasting of pistachios produces less acrylamide (119 µg/kg), than during the IR method (318 µg/kg), or using hot air (204 µg/g) (Table 2). The effect of microwave frying on acrylamide formation was also investigated in the coating part of chicken. Microwave frying resulted in lighter colored samples and lower acrylamide formation in the coatings prepared with different types of flours comparing to the conventional frying [92]. The highest reduction (34.5%) in acrylamide level was observed for batter containing rice. Some authors also reported that short exposure to microwaves (blanching and thawing) may even limit the formation of acrylamide during the final heat treatment. These operations reduce frying time and hence less acrylamide is formed [78,95]. In the study of Akkarachaneeyakorn et al. [96], combined microwave–hot air was an effective method for the production of coffee, chocolate, and black malt with reduced acrylamide when compared with the conventional roasting process. This reduction was a consequence of shortened total roasting time. According to the authors, such a combined roasting process can be potentially applied to products such as roasted coffee, which is usually processed at high temperature for an extended period to reduce AA formation. Al-Ansi et al. [78] confirmed the beneficial effect of microwave treatment on the reduction of acrylamide content. They compared the level of this compound in traditional (190 °C/10 min) and microwave (700 W/90 s) baked biscuits and determined that 10% less acrylamide content was found in microwave baked ones. The authors observed a greater acrylamide reduction after adding to the dough different amounts of black cumin seeds, which resulted in AA decrease from 17% to 53% in traditionally baked biscuits and from 23% to 68% in microwave baking. According to the authors, such a large acrylamide reduction was caused by the antioxidants found in black cumin seeds, which confirmed the high, statistically significant negative values of the correlation coefficients between the level of acrylamide and the antioxidant activity measured by various methods.

It seems that contradictory results on the effect of microwave heating on the AA formation in food may be mainly due to differences in microwave heating parameters (microwave power, heating time) used in the various tests, as well as the chemical composition and type of heated food, including water activity levels. These polar particles of food ingredients, including water molecules, are subjected to intense electromagnetic field action, resulting in heat energy generation inside the product and its temperature increase, but also in accelerating the reaction between food ingredients. The type and intensity of interactions between compounds is largely determined by the electromagnetic field strength, its frequency, wave type, modulation, and exposure time. Most authors reported that with the increase of the microwave heating power, the acrylamide content increases [35,90,94,97]. Sansano et al. [94] indicated that it depends on the treatment time, because in the case of a short processing (1–5 min), the acrylamide content decreases with increasing microwave heating power. In the research carried out on microwave-fried potatoes strips, the power increase from 430 W to 600 W resulted in a nearly two-fold reduction in AA content. The authors argued that it may be due to the protector effect of the steam flow from the center of the samples dragging both the acrylamide formed and its precursors. When increasing the power of microwave heating for more than 5 min, the acrylamide content increased, and when the process was extended further, it decreased, because degradation of this compound was probably occurring. Elfaitouri et al. [97] reported, that the increase of the power used during frying potato chips from 200 W to 800 W (with the temperature increase from 160 °C to 180 °C) increases the acrylamide content several times. Although they did not show such clear trends associated with duration of the frying process, the relationship of increasing acrylamide content with increasing the time of microwave food processing was confirmed by other authors [12,35,83,89]. Chen et al. [90] reported that with the extension of the microwave heating time of shrimp chips from 40 s to 80 s, the acrylamide content increased almost 20 times (the process was carried out at 900 W).

Yuan et al. [35] suggest, that microwave power for carbohydrate-rich food processing should be used at the lowest possible level. They reported that, in comparison to conventional heating, such as boiling or frying, strong microwave treatment is more favorable for the AA formation in both asparagine/fructose and asparagine/glucose model systems and in potato chips. Zyzak et al. [88] in order to confirm the importance of this mechanism of acrylamide formation tested what amount of this compound will be formed after enzymatic decomposition of asparagine. For this purpose, they added the asparaginase (which catalyzes the hydrolysis of asparagine into aspartic acid and ammonia) to microwaved mashed potato slurry used for snacks production. These studies indicated that there was more than 99% reduction in the amount of acrylamide formed in the samples with the addition of the enzyme, compared to the samples prepared in the same way but without the enzyme. That confirmed the major role of asparagine and a reactive carbonyl group in acrylamide formation.

Yuan et al. [35] determined that the formation of acrylamide during the microwave heating of potato chips was significantly dependent on the pH value. At pH 3.0, this compound was not detected at all, but as the heating time was extended and the pH value was increased from 4.0 to 8.0, the AA content increased intensively. Heating for 5 min and longer resulted in almost 3 times faster formation of this compound at pH 8.0, compared to pH 4.0. With the lack of acrylamide generation at pH 3.0, the authors explain that most groups of asparagine and the ring of sugar occurred in their protonated form, preventing reaction between asparagine and sugar.

Sansano et al. [94] emphasize the effect of the water level in food on the formation of acrylamide. The authors claim that, as the parameters of microwave processing increase, the protection of the water flow gets lost and acrylamide content increases considerably. It was shown, that keeping the product surface moist highly limits the acrylamide formation in baked or fried products [98]. On the other hand, a jet of vaporized water from the surface of the product to the external oil during frying would sweep away some of the acrylamide produced, which is a very unstable and volatile compound, as well as its precursors. This volatilization phenomenon is exacerbated when microwave power is applied due to the volumetric heating and the larger water flow [94].

In addition to the use of the enzyme and the black cumin seeds mentioned above, it is advised to also use other compounds that can reduce the content of acrylamide formed during food processing. Asadi et al. [12] indicated that the addition of salt affects the formation of acrylamide in microwave-heated pistachios (Table 2). Microwave roasting of salted pistachios resulted in the formation of smaller amounts of acrylamide, compared to unsalted products roasted at the same time, which, according to the authors, was related to their water content. For salting, the nuts were soaked in water with 20% NaCl addition, which meant that part of the roasting time has been spent for drying the pistachios. The importance of salt added to food products, as well as other additives of various types (e.g., nicotinic acid, citric acid, or glycine) in reducing acrylamide content in various types of heated food was also emphasized by Maan et al. [6]. Chen et al. [90] analyzed the acrylamide content of microwave-puffed shrimp chips fortified with different amounts of calcium salts and indicated that the greatest mitigation of acrylamide formation was obtained with the addition of 0.1% calcium lactate. This could be due to the fact that adding calcium ions enhanced the stability of asparagine-matrix interactions at high temperatures, thus preventing asparagine from reacting with carbonyl precursors to form acrylamide during Maillard reactions.

Another component lowering the acrylamide content may be phenolic compounds [11,40], which have strong antioxidant properties. Zhang et al. [99] investigated the-dose response effect in a potato-based model system with the addition of asparagine, glucose, and 24 selected flavonoids using microwave heating. They indicated that all flavonoid fractions inhibit acrylamide formation, and some of them can reduce the formation of this compound by up to 91.9%. The authors explained this action by the fact that phenolics may compete with the carbonyl groups of reducing sugars in Maillard reactions and then affect the generation of acrylamide. The degree of the acrylamide reduction by phenolic compounds showed a close correlation between the number of phenolic hydroxyls instead of alcoholic hydroxyls in different types of flavonoids. Soncu and Kolsarici [100] also attributed a beneficial effect of phenolic compounds in reducing AA content, recommending the addition of green tea extracts containing large amounts of these compounds to poultry products. They claim that the radical-trapping effect of catechins or inhibition of the Maillard reaction by catechins present in green tea could be considered a possible explanation for the reduction of acrylamide formation.

Elfaitouri et al. [97] investigated the effect of fat used for frying and showed that using the one with greater oxidative stability results in the formation of less acrylamide. The importance of the freshness of the frying oil and the formation of less acrylamide with the use of fresh oil was also confirmed by Lee et al. [10]. The authors concluded that, regardless of the freshness of the fat used for frying, it is most advantageous not to use oil at all because of the acrylamide formation.

Summarizing the results of research given in the literature, it can be concluded that microwave blanching or pre-thawing of frozen foods, in contrast to other thermal microwave food processing, do not cause significant changes in the formation of acrylamide. Short processing time, low temperature of heating, and relatively low power level of microwave oven in microwave blanching and thawing or even initial microwave heating are advantageous to reduce quality losses, especially for reducing acrylamide formation. Therefore, short exposure to microwaves can be recommended to minimize the undesirable changes in flavor and texture, as well as nutrient losses and to reduce the amount of acrylamide formation. However, in contrast to the above-mentioned processes, in some microwave-heated foods products, acrylamide can be easily generated when more radical treatment parameters are used.

There are many mechanisms of microwaves’ influence on food ingredients. In 2005, Rydberg et al. [101] in the study of the effect of microwave heating on mashed potato, showed that the formation of acrylamide progresses with the pyrolysis of the sample. In this experiment, the increase of the acrylamide content was 140 times higher after prolonging the heating time from 100 s to 150 s. Microwave heating gave rise to uncontrollable variations, mainly due to inhomogeneous heating resulting in local pyrolysis. Other studies suggested that AA in food is generated from pyrolytic fragments of asparagine and that this reaction is facilitated by concomitant pyrolysis of Maillard-active dicarbonyl and hydroxycarbonyl precursors [102]. Fernández et al. [66] also reported that microwave heating favors the pyrolysis process. This leads to the conclusion that, unlike conventional methods, microwave heating can generate the acrylamide formation, largely due to intense simultaneous pyrolysis of AA precursors in Maillard reactions. The intensity of the pyrolysis process is probably related to the electromagnetic field strength, its frequency, wave type, modulation, and exposure time. In addition, recent research has provided more evidence for the specific effects of microwaves, e.g., on the biological structures of microbial cells, which have so far been frequently considered as the effects of a thermal factor [103,104]. Therefore, it appears that changes in food under microwave heating, including the AA formation, should not be considered only as a result of the thermal factor, but also other factors that cause intensification of changes in food. Since the mechanism of these reactions is not clear, further research is needed to better understand the formation of acrylamide during microwave heating. This is important because it seems that the use of microwave heating in the future will increase both for industrial food preservation and domestic heating as a common way of preparing foods at home.

One of the important differences between conventional and microwave heating is that the first one causes an intense browning of the product surface and the other does not [105,106]. Brown surfaces, produced by the Maillard reaction and caramelization of sugars during traditional heating, are a result of high temperatures accompanied by dehydration [101,102]. In the case of microwave heating, the cool ambient temperature inside a microwave oven causes surface cooling of microwave-baked products and low surface temperature prevents Maillard browning reactions from being formed. As a result, microwave products are not as tasty as those which are traditionally baked. Many researchers have associated the browning intensity with the AA content in many food products such as fried potatoes and bread, including crisp bread, coffee, biscuits [16,46,67,107,108,109,110,111,112,113]. According to some authors, the browning of food products during technological processing may be used as an indicator of acrylamide content [17,114,115]. Surdyk et al. [113] reported that the acrylamide formation is a surface reaction and over 99% of AA is formed in a surface layer of food heated by conventional methods—e.g., after baking in bread crust and only 1% in crumb—showing a significant correlation between surface color and the acrylamide content in the crust. The surface of the microwave-heated food remains moist and the development of the crust, the taste, the color and also the AA formation is limited [85,86]. Although surface browning is a phenomenon typical of roasted and fried foods, and correlates with acrylamide levels, many studies showed, however, that microwave heating, which does not cause more browning of food, can result in more acrylamide formation than conventional heating [15,35,85,86,87]. It can be concluded that, unlike conventional heating, microwave heating can cause acrylamide formation throughout the entire product without its intensive generation in the surface layer. This results in a higher content of this toxic compound in the heated product, without the significant browning increase of its surface [15]. Similar observations were made also by Erdoğdu et al. [95]. As mentioned earlier, the mechanism of these reactions is unclear and further studies are needed to better understand the acrylamide formation during microwave heating.

## 5. Conclusions

On the basis of the available data, it should be concluded that the formation of acrylamide in microwave-heated food, as in the case of conventional heating, depends on the process parameters and the properties of the processed products. The main advantage of microwave heating is the short duration of this process, but due to the possibility of uncontrolled and uneven temperature increase in individual layers of the product, which favors a greater amount of acrylamide forming, further research in this area should be carried out.

It is important to find a compromise between the amount of AA generated and the sensory characteristics of the microwave-heated food. Additional substances, such as black cumin or tea extracts, can help in this matter, because their strong antioxidant properties reduce the formation of acrylamide and at the same time give food new and interesting sensory properties. More attention should be focused on the influence of microwave heating on such sensory properties of the products, like the browning and the formation of flavor compounds creating distinctive tastes and aromas that are much desired by consumers.

Theoretical deliberations based on the mechanism of action of microwave radiation and the expected amount of formed acrylamide are not always reflected in the actual course of this process. It seems that due to the formation of AA, the lowest possible microwave heating power and the shortest duration of this process should be used, which can be applied—especially in pre-cooking.

In particular, domestic use of microwave ovens requires control and the creation of appropriate recommendations for consumers who use them, because the industrial use of microwave energy is much easier to monitor. Consumers need clear guidelines on the used technique for food preparation, in a way which minimizes the risk of formation of excessive harmful compounds, including acrylamide.

## Figures and Tables

**Figure 1 molecules-25-04140-f001:**
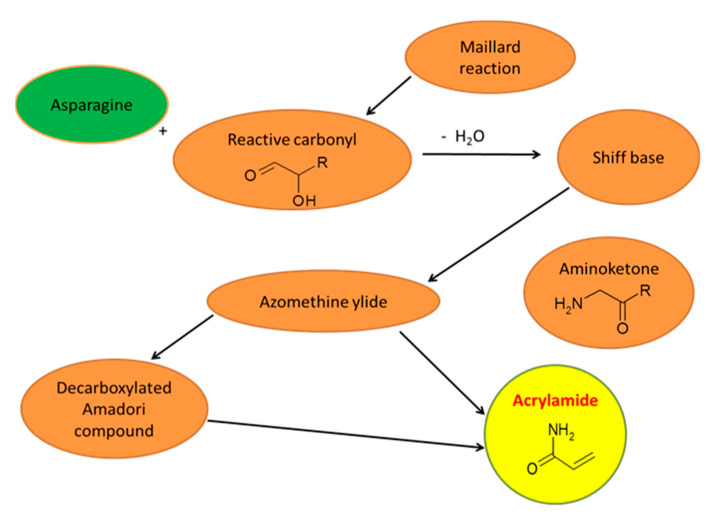
Simplified mechanism of acrylamide formation by Maillard reaction.

**Table 1 molecules-25-04140-t001:** Acrylamide content in various food products (µg/kg)

Food Product	µg/kg	References
**Baby foods**		
Cereal-based (ready-to-eat)	13	[47]
Instant cereal based	345	[47]
Candy-bars	54	[47]
Biscuits	87	[32]
Jarred baby food	32–47	[48]
Ready-to-eat meal cereal-based	13	[49]
Porridge	29	[49]
Infant formulae	14	[49]
Fruit purée	22	[49]
Juice	12	[49]
**Bread**		
Crisp bread	443	[32]
Wheat soft bread	38	[49]
Other soft bread	57	[49]
**Cereal products**		
Wheat- and rye-based products	170	[49]
Bran products and whole grain cereals	211	[49]
Crackers	231	[49]
Biscuits and wafers	201	[49]
Gingerbread	407	[49]
Pasta	13	[49]
Beer	14	[49]
**Cacao**		
Cacao (100% cocoa powder)	347	[32]
Cacao (cocoa-containing beverages powder: sugars and 20% cocoa powder)	248	
**Coffee and coffee substitutes (dry)**		
Roasted coffee (dry)	249	[49]
Instant coffee (dry)	710	[49]
Substitute coffee (dry), based on cereals	510	[49]
Substitute coffee (dry), based on chicory	2942	[49]
**Potato products**		
French fries	326–328	[48]
Potato crisps	689–693	[48]
Deep fried home-cooked potato products	234–241	[48]
Oven baked home-cooked potato products	317	[48]
**Other products**		
Roasted nuts and seeds	93	[49]
Black olives in brine	454	[49]
Prunes and dates	89	[49]
Paprika powder	379	[49]
Fish and sea food	25	[50]
Milk and milk products	6	[50]
Pizza	33	[50]
Green tea roasted	306	[50]
Sugars and honey	24	[50]
Vegetables	17	[50]
Vegetable crisps	1846	[49]
Fruits dried and processed	131	[50]
Dried food	121	[50]

**Table 2 molecules-25-04140-t002:** Effect of conventional and microwave processing techniques on acrylamide content in different food products (µg/kg).

Food Product	Preparation Method		Acrylamide (µg/kg)	References
French fries	Before final preparation		416	[84]
	Pan frying 180 °C/3 min		561	
	Deep frying 180 °C/3 min		597	
	Roasting 220 °C/10 min		727	
	Microwaving 220 °C (700 W)/10 min		790	
	Deep frying 180 °C/from 1 to 8 min		21–231	[94]
	Microwave frying 315 W/from 1 to 10 min		46–182	
	Microwave frying 430 W/from 1 to 8 min		44–337	
	Microwave frying 600 W/from 1 to 6 min		23–172	
	Unthawed and deep frying at 180 °C/3.5 min		85	[82]
	Thawing at room temp. and deep frying at 180 °C/3.5 min		84	
	Thawing in a chiller (5 °C overnight) and deep frying at 180 °C/3.5 min		77	
	Thawing in a microwave oven (30% power/5 min) and deep frying at 180 °C/3.5 min		106	
Potato pancakes	Before final preparation		286	[84]
	Pan frying 180 °C/3 min		437	
	Deep frying 180 °C/3 min		422	
	Roasting 220 °C/10 min		564	
	Microwaving 220 °C (700 W)/10 min		694	
Potato chips	Frying 180 °C/4 min		645	[35]
	Microwaving 750 W/2.5 min		897	
	Frying 160 °C/7 min		3110	[87]
	Frying 180 °C/6 min		3604	
	Microwaving 750 W/3 min		5184	
	Microwave frying at 160 °C (200 W)/30–150 s		542–895	[97]
	Microwave frying at 170 °C (400 W)/30–150 s		669–1739	
	Microwave frying at 180 °C (800 W)/30–150 s		1139–11,423	
Grated potatoes	Frying		447	[33]
	Microwaving		551	
Potato	Baking 220 °C/5 min		approx.70	[86]
	Microwave precooking 150 W/60 s and baking (220 °C/5 min)		approx. 180	
Asparagus	Baking 220 °C/5 min		approx. 90	
	Microwave precooking 150 W/60 s and baking 220 °C/5 min		approx. 160	
Green gram sprouts	Baking 220 °C/5 min		approx. 340	
	Microwave precooking 150 W/60 s and baking 220 °C/5 min		approx. 580	
Pistachios	Raw		57	[12]
	Sun-dried		93	
	Hot air roasting from 100 °C/5min to 150 °C/5min	salted	130–463	
		unsalted	204–594	
	IR (infrared) roasting from 75 V/10 min to 95 V/30 min	salted	242–697	
		unsalted	318–851	
	Microwaving roasting form 180 W/12 min to 360 W/16 min	salted	105–307	
		unsalted	119–344	
Pre-cooked flour-based croquettes	Before final preparation		190	[15]
	Pan frying 180 °C/5 min		285	
	Deep frying 180 °C/5min		298	
	Roasting 200 °C/10 min		360	
	Microwaving 200 °C (700 W)/10 min		420	
Fried chicken with batter formulation of chickpea flour	Deep frying (180 °C/5 min)		110	[92]
	Microwave frying (180 °C/350 W/2 min)		79	
Fried chicken with batter formulation of rice flour	Deep frying (180 °C/5 min)		111	
	Microwave frying (180 °C/350 W/2 min)		73	
Fried chicken with batter formulation of soy flour	Deep frying (180 °C/5 min)		100	
	Microwave frying (180 °C/350 W/2 min)		76	

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
