# Peer review of "Effect of Microwave Heating on the Acrylamide Formation in Foods"

_molecules, 2020, doi:10.3390/molecules25184140_

Round 1
Reviewer 1 Report
In this review, the authors investigated the influence of microwave heating technology on the formation of acrylamide in food. Through summarizing the previous studies, the authors concluded that Microwave heating at a high power level could cause a greater AA formation while using microwave heating at low power may reduce the formation of AA. Moreover, the authors conjectured this phenomenon might be related to the electromagnetic field on food.
Overall, the entry point of the literature is OK, and the idea is clear, although there are still some deficiencies and confusing issues which need to be further addressed.
Major concerns:
1. In this study, the authors demonstrated that microwave heating could affect the formation mechanism of acrylamide. However, the explains or conjectures of the specific formation mechanism were not enough.
2. In Chapter 4 “Acrylamide in microwave heating”, the authors mentioned that microwave heating at high or low power might cause opposite effect on formation of AA and attribute to different parameters. But only the influence of electromagnetic field was concluded, as to other parameters including time, temperature and humidity of food were not concluded.
3. In the first half of the article, the author introduced characteristics and application of microwave heating, did different applications have different effects on formation of acrylamide in food?
Minor concerns
1. Line 263, “as shown”, the reference was ambiguous.
2. Line 363-367, the authors mentioned a possible explanation, did this explanation mean that a rapid temperature rise resulted in more AA formation?
3. Conclusions: The microwave heating of high/low power, the specific data or the range?
Author Response
We place responses to the review in a separate file.

Reviewer 2 Report
Review of Manuscript ID: molecules-872522 Title: Microwave application in food industry and its impact on the acrylamide formation in foods.
Authors: Joanna Michalak, Marta Czarnowska-Kujawska, Elżbieta Gujska, Joanna Klepacka
The publication presented for review is original and interesting. The work summarizes and collects the current state of knowledge in the field of harmfulness and formation of AA and microwave technology in its formation. This work may be an important source material for food technologists. It is written in clear and understandable language. In my opinion, the publication will significantly increase in value if the authors pay attention to and consider my following observations:
- in chapter 2 entitled "the structure and properties of acrylamide" I would suggest that the authors add information on the influence of the quantity and mutual ratio of precursor substances on AA formation. It would also be worth to include in the table the amounts of AA found in various food products. In addition, it would be worth providing information on how to protect the human body against AA, e.g. through bioactive substances.
- in subsections of chapter 3 (blanching, drying, thawing and tempering, pasteurization, cooking and baking) it would be worthwhile to specify how the use of microwave heating in these processes affects the formation of AA in the final product.
- while the information contained in chapter 4 entitled "arcylamide in microwave heating" would be good to summarize or collect in the form of a table or diagram that would show how the raw materials and their processing parameters affect the formation of AA in food. In my opinion that would be a great value of work. It won't be easy, but it's worth doing. It will be very important.
Author Response
We place responses to the review in a separate file

Reviewer 3 Report
The authors in this review present the application of microwaves in different procedures in the food industry, discussing the advantages and disadvantages. The influence of microwave thermal food processing on the formation of Acrylamide was presented based on the published studies. The manuscript is well written and covers the available literature on the topic. The authors concluded that microwave heating may be not a safe food treatment because at high power level greater amounts of Acrylamide can be formed. Furthermore, the research should be continued in the application of microwaves in the food industry. Minor revision is needed to be published.
Specific comments:
- Line 145. Reference 50 is a Commission Regulation not a Recommendation.
- Correct reference 51 with Commission Recommendation 2019/1888/EU.
- The Commission Recommendation 2019/1888/EU is about the monitoring of other foodstuff that were not covered by the previous Regulation and Recommendations.
- Line 147. Correct with “…in 2013 [49] and 2017 [50]….”
- Section 3.4 should be Pasteurization and Sterilization because there is also discussion about sterilization. In this section the authors suggest that although there are many limitations, microwave pasteurization was successfully commercialized, but the use of microwaves in sterilization has limited success in commercialization. Is that correct?
- Is reference 67 relevant in Section 3.5?
Author Response
We place responses to the review in a separate file (as attachment).

Reviewer 4 Report
Acrylamide formation in some thermal process is a highlight topic, since several countries are establishing limits, due to its carcinogenic nature. I leave some comments for the authors in the attached file but I would like to stress some points:
- In my opinion the paper should be focus on the effect of microwave heating rather than application of microwave in food industry, and it should be reflected in the title. I would propose the following title: Effect of microwave heating on the acrylamide formation in foods.
- In this sense, section 3 should be deeply revised and reduced. It is more for a book´s chapter, not for a review.
- Section 4 should be the main part of the review and should be amplified. I think that there are more references about the topic to be considered, and above all, more recent references. The recent references have been cited in the introduction, but I miss recent references in the section 4 (from the last 5 years).
- Conclusions should be reformulated and I would expect a strong conclusion of the analyzed papers. It seems that AA formation when microwaves are applied depends on the work.

Author Response
We place our responses to the review in a separate file (as attachment).

Round 2
Reviewer 4 Report
I would like to thank the authors their kindly answers to my questions and I appreciate your deep revision of the paper. In my opinion the paper has greatly improved and accomplish the high standards of the journal.